# ENHANCING SEMI-SUPERVISED LEARNING VIA SELF-INTERESTED COALITIONAL LEARNING

## ABSTRACT

Semi-supervised learning holds great promise for many real-world applications, due to its ability to leverage both unlabeled and expensive labeled data. However, most semi-supervised learning algorithms still heavily rely on the limited labeled data to infer and utilize the hidden information from unlabeled data. We note that any semi-supervised learning task under the self-training paradigm also hides an auxiliary task of discriminating label observability. Jointly solving these two tasks allows full utilization of information from both labeled and unlabeled data, thus alleviating the problem of over-reliance on labeled data. This naturally leads to a new learning framework, which we call Self-interested Coalitional Learning (SCL). The key idea of SCL is to construct a semi-cooperative "game", which forges cooperation between a main self-interested semi-supervised learning task and a companion task that infers label observability to facilitate main task training. We show with theoretical deduction its connection to loss reweighting on noisy labels. Through comprehensive evaluation on both classification and regression tasks, we show that SCL can consistently enhance the performance of semi-supervised learning algorithms.

## 1 INTRODUCTION

Despite the huge success of supervised learning with deep neural networks, they require large amounts of labeled data for model training to achieve high performance (Meng et al., 2019; Prabhu & Varma, 2014). Collecting labeled data can be very difficult and costly in practice (Meng et al., 2020). The challenge can be further exacerbated as in many cases, human or machine labeled data may contain errors or noises (Natarajan et al., 2017; Schnabel et al., 2016). Semi-supervised learning (SSL) (Chapelle et al., 2006) that leverages both labeled and unlabeled data has become an increasingly promising yet still challenging area, and is widely applicable to real-world problems.

To utilize the hidden information in the unlabeled data, a general and popular semi-supervised learning paradigm is through self-training (Yarowsky, 1995; Lee et al., 2013). Self-training uses a previously learned model to predict labels for the unlabeled data (pseudo-labeling) which are then used for subsequent model training (Yarowsky, 1995; Lee et al., 2013; Laine & Aila, 2017; Tarvainen & Valpola, 2017b; Iscen et al., 2019b). Despite some empirical successes, self-training methods still suffer from two core challenges, which are *over-reliance on labeled data* and *error accumulation*. Most existing semi-supervised learning methods (Yarowsky, 1995; Lee et al., 2013; Laine & Aila, 2017; Sohn et al., 2020; Xie et al., 2020; Kipf & Welling, 2017) assume the labeled and unlabeled data follow the same or similar data distribution, thus the data-label mapping mechanisms learned from the labeled data are somewhat transferable to unlabeled data. Essentially, the algorithm is still reinforcing the information in the labeled data rather than mining additional information from the unlabeled data. Second, the pseudo-labels of unlabeled data can be incorrectly predicted. Using such biased information for training in subsequent epochs could increase confidence in erroneous predictions, and eventually leading to a vicious circle of error accumulation (Cai et al., 2013; Arazo et al., 2020). The situation can be even worse when the labeled data contain errors or noises, as the model learned from labeled data will make more mistakes, resulting in more severe error accumulation.

Alleviating the issue of over-reliance on labeled data requires extracting additional sources of information from the unlabeled data. One recent approach is to introduce input consistency regularization (Sajjadi et al., 2016). Consistency regularization enforces the stability of the predictions with respect

to the transformations of the unlabeled data (e.g. data augmentation on images such as rotation, flip and shift, etc.). Although such data perturbation or augmentation are well-defined for images, they are not directly transferable to general settings, hence the success of consistency regularization methods are mostly restricted to image classification tasks (Sajjadi et al., 2016; Liu et al., 2019; Xie et al., 2020; Sohn et al., 2020). We provide a new insight by noting that, under the self-training paradigm, every semi-supervised learning task hides another auxiliary task of discriminating whether the data label is real or a pseudo-label predicted by a machine oracle. Jointly solving the main task together with the auxiliary task allows sufficient utilization of the hidden information in unlabeled data.

To break the vicious circle of error accumulation, one needs to not always trust the machine-labeled data. A natural idea is to consider the reliability or confidence of data labels and treat them differently during training. A connected approach in supervised learning literature is to use the notion of "soft label" with confidence probability or weights, which has been verified in a number of noisy label learning problems (Natarajan et al., 2017; Schnabel et al., 2016; Shen et al., 2020; Algan & Ulusoy, 2021). This treatment can also help a semi-supervised learning algorithm gain the ability to robustly learn from noisy labels.

Combining previous insights, we develop a novel semi-supervised learning framework, called Self-interested Coalitional Learning (SCL). SCL contains three ingredients. First, it simultaneously solves a companion task by learning a discriminator as a critic to predict label observability (whether it is a true label or a pseudo-label), which is used to mine additional information in the unlabeled data while facilitating main task learning. Second, the output of the discriminator is in fact a measure of label reliability, thus can be interpreted as the confidence probabilities of labels (Qin et al., 2021). This converts the original problem into a soft label learning problem. Finally and most importantly, SCL introduces a cooperative-yet-competitive learning scheme to boost the performance of both the main task and the companion task. Specifically, SCL forges cooperation between both tasks by providing extra information to each other (the main task provides prediction loss, companion task provides label confidence measures), leading to improved performance of both tasks. Moreover, we consider the main task model to be self-interested which tries to challenge the discriminator by providing as little information as possible. This design forms a semi-cooperative "game" with a partially adversarial main task model, which can be shown theoretically equivalent to a loss reweighting mechanism on noisy soft labels. Under this design, the main task and the companion task in SCL are neither fully cooperative as in multi-objective optimization (Deb, 2014) nor fully adversarial to each other as in adversarial learning (Goodfellow et al., 2014; Matyasko & Chau, 2018), but balance the benefit of cooperation and competition. In summary, the contribution of this work include:

- We develop a novel and very general self-interested coalitional learning framework for SSL that fully utilizes the hidden information in the unlabeled data while addressing the error accumulation issue in the self-training paradigm of SSL.
- SCL can be easily incorporated into a wide range of semi-supervised algorithms (e.g. image classification, label propagation and data imputation) with limited changes on the original method. We show with comprehensive empirical experiments that SCL consistently achieves superior performance and robustness compared with the original and self-training version of different semi-supervised learning algorithms.
- SCL naturally provides the data confidence measures from the discriminator as a byproduct of the learning process, which offers additional interpretability of the semi-supervised learning task. This can be particularly useful for many practical applications, while also providing extra benefits for scenarios with noisy data labels.

## 2 Preliminaries

### 2.1 Problem definitions and formulations

We formulate the semi-supervised learning problem as learning a model $f(\cdot)$ with input data $x \in X$ to predict the label $y \in Y$. In SSL, only a partial set of data labels $Y_L$ with ground truth are given in the training set, the rest are unlabeled $Y_U$ ($Y = Y_L \bigcup Y_U$). For convenience of later discussion, we denote $L$ as the set of indices of data samples with labels and $U$ as the set of indices for unlabeled data. In many real-world SSL problems, the size of labeled data $|L|$ is often limited. In extreme cases, $Y_L$ may potentially contain errors or noises. We further define a mask vector $M$ to denote the

observability of labels over each data sample. We set $M_i \in M$ equals to 1 if $i \in L$ and $M_j \in M$ equals to 0 if $j \in U$. In our proposed SCL framework, we consider following common settings of semi-supervised classification (inductive classification, label propagation) and regression (data imputation) tasks:

**Inductive classification:** a typical SSL task setting is to construct a classifier to predict labels $y \in Y$ for any object in the input space $x \in X$. Common examples include semi-supervised image classification tasks (Sajjadi et al., 2016; Sohn et al., 2020; Xie et al., 2020), where only a subset of data samples are given the known labels $Y_L$ and the rest data labels $Y_U$ are unknown.

**Label propagation:** another class of SSL tasks under transductive setup is to use all input samples $X$ and observed labels $Y_L$ to train a classifier to predict on unseen labels $Y_U$. Examples include classifying nodes in a graph given only small subset of node labels $Y_L$ (Kipf & Welling, 2017).

**Data imputation:** one of the special case of SSL problem is that the missing state of input and output data are strongly correlated (Richardson et al., 2020; Qin et al., 2021), such as the data imputation tasks. In data imputation, the inputs $X$ are partially filled and the labels $Y$ are equivalent to a reconstructed version of $X$ obtained through regression.

## 2.2 AUXILIARY TASK IN SEMI-SUPERVISED LEARNING PROBLEMS

We begin our discussion by first noting that, if we feed the unknown labels $Y_U$ with the model $f$ predicted labels $\tilde{Y}_U = \hat{Y}_U^{(k)}$ ($\hat{Y}_U^{(k)} = f(X)$ represents the labels from the $k$-th round of pseudo-labeling) or other machine-generated labels, there actually hides an auxiliary task of discriminating whether the data label is real or a pseudo-label. Denote all the data labels under pseudo-labeling as $\tilde{Y} = Y_L \bigcup \tilde{Y}_U$. We can train a discriminator $d(\cdot)$ to tell the confidence measure $p \in P$ of whether $x \to \tilde{y}$ ($x \in X, \tilde{y} \in \tilde{Y}$) is a valid mapping. This is always learnable since the ground truth of $P$ is exactly the observability mask $M$. With a slight abuse of notation, we formulate the main SSL task A and the auxiliary companion task B as follows:

$$
\begin{aligned}
A : f(X) = \hat{Y}, & \qquad \mathcal{L}_A = loss_A(Y_L, \hat{Y}_L) \\
B : d(X, \tilde{Y}) = P, & \qquad \mathcal{L}_B = loss_B(P, M)
\end{aligned}
\tag{1}
$$

where $loss_A$ is the original loss function of the main task and $loss_B$ can be any binary classification loss between $P$ and $M$, such as the binary cross entropy (BCE) loss. Jointly solving the above two tasks allows exploiting the underlying relationship between input data $X$ and data label $Y$ from another angle, which can potentially provide more information to facilitate main task training.

There are some other works that embody the similar idea of jointly learning two tasks. For example, the multi-task style multi-objective optimization methods (Ruder, 2017; Deb, 2014) exploit the shared information and underlying commonalities between two tasks and solve the problem by minimizing an augmented loss. The generative adversarial learning (Goodfellow et al., 2014) makes a generator and a discriminator learn against each other, thus improves the performance of both tasks. However, naïvely applying these two approaches has some drawbacks. For example, multi-objective optimization cannot handle potential contradictions of the two tasks in certain settings, where jointly minimizing the augmented loss may impede both tasks from achieving the best performance (Qin et al., 2021). Whereas under the adversarial learning framework, $loss_A$ is implicitly optimized through the discriminator, which could result in potential loss of information. Moreover, solving the minimax optimization problem under a GAN-style model is much harder compared with directly minimize both loss functions explicitly in (1) in a supervised learning fashion.

## 3 SELF-INTERESTED COALITIONAL LEARNING

In this section, we introduce a new learning strategy to jointly solving the two tasks in Problem (1) for semi-supervised learning, which is called self-interested coalitional learning (SCL). As its name suggests, SCL forges cooperation between a self-interested main task and an unselfish companion task for boosted performance. We show with theoretical derivation that under the pseudo-labeling paradigm, this special learning scheme will lead to a soft label learning formulation for the original SSL task. We also provide an intuitive interpretation of the SCL scheme.

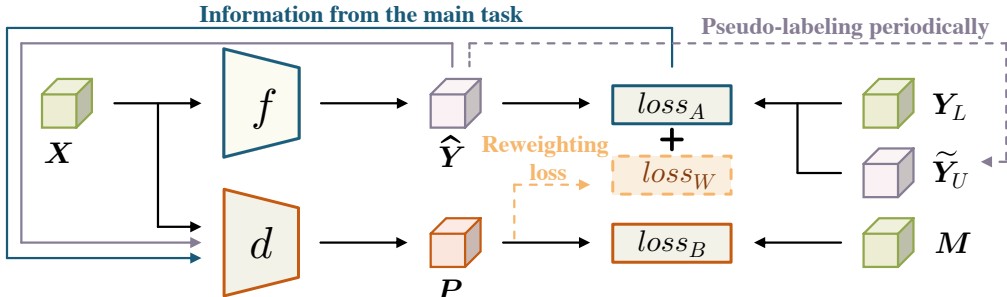

Figure 1: Illustration of self-interested coalitional learning

## 3.1 ALTERNATIVE FORMULATION UNDER SCL

We consider an alternative formulation of Problem (1) by modifying the loss of the main task $\mathcal{L}_A$ and the inputs of the companion task to establish cooperation and information sharing:

$$
\begin{aligned}
A &: f(X) = \hat{Y}, & \mathcal{L}_A &= loss_A(\tilde{Y}, \hat{Y}) + \alpha \cdot loss_W(\tilde{Y}, \hat{Y}, P) \\
B &: d(X, \hat{Y}, g(\tilde{Y}, \hat{Y})) = P, & \mathcal{L}_B &= loss_B(P, M).
\end{aligned}
\tag{2}
$$

where $g(\tilde{y}, \hat{y})$ is some form of information provided by the main task. We can simply model the information $g(\cdot)$ as the element-wise loss term of the main task, i.e. $g(\tilde{y}, \hat{y}) = loss_A(\tilde{y}, \hat{y})$, such as the cross entropy loss for classification tasks and mean square error (MSE) loss for regression tasks. Moreover, $loss_W(\tilde{Y}, \hat{Y}, P)$ is a corrective additional loss term impacted by the label confidence measure information $P$ provided by the discriminator $d$, and $\alpha$ is its weight parameter. Due to the existence of the corrective information from $loss_W$, we now train the main task with all data labels $\tilde{Y}$ rather than only with observed labels $Y_L$, thus allows sufficient utilization of information from the unlabeled data. We illustrate the strategy of SCL in Figure 1.

The design of SCL has several interesting properties. First, since additional information are provided to both tasks, they can be both learned better. Second, a less well-learned model $f$ will produce large element-wise error $loss_A(\tilde{y}, \hat{y})$ for a predicted label $\hat{y}$, which in turn provides more information in $g(\cdot)$ for the discriminator $d$ to distinguish whether a label is real or a pseudo-label. As $f$ is learned better during the cooperative learning process, when $g$ no longer provides useful information for discriminating the label observability, then we may have reason to believe $f$ has achieved satisfactory inference performance. Hence minimizing the information in $g(\cdot)$ coincides with improving the learning of $f$. This suggests that if we alter the learning direction of $f$ using $loss_W$ to encourage it providing as little information in $g(\cdot)$ as possible, then we might obtain a better learned $f$. This in fact forms a cooperative-yet-competitive "game" between both tasks, with the main task been self-interested and partially competitive. Finally, it also allows the discriminator to detect errors or noises in observed data labels. As the erroneous $(x, y)$ data pairs may possess different $x \rightarrow y$ mapping patterns against normal data, thus are more likely to be identified as pseudo-labels. Such negative impact can be further corrected using $loss_W$ with label confidence measure $P$.

## 3.2 DERIVATION OF THE ADDITIONAL LOSS TERM $loss_W$

Until this point, the only mystery about SCL is the exact form of $loss_W$ that embodies the self-interested behavior for the main task model $f$. We resort to calculus of variations to provide theoretical derivation of $loss_W$.

Under the alternative formulation (2), both the discriminator $d$ and its loss $\mathcal{L}_B$ are impacted by the information $g(\cdot)$ provided by $f$, thus they are now functionals of $f$. We wish to inspect how the variation of model $f$ impacts $\mathcal{L}_B$. Denote $\theta_f, \theta_d$ as the model parameters of model $f$ and $d$. Here, we fix the the discriminator $d$ and its parameters $\theta_d$ to exclude its own influence, then $d(\theta_d, x, f)$ can be replaced by $d(x, f)$. Similarly, with $d$ fixed, we can define the functional $J(f)$ for $\mathcal{L}_B$ and further eliminate the effect of changes in $x$ on $d$ by integrating over $x$, mathematically:

$$
J(f) = \mathcal{L}_B = \int \frac{\partial \mathcal{L}_B}{\partial d} dx = \int F(x, f, f') dx
\tag{3}
$$

where $F(x, f, f') = \partial \mathcal{L}_B / \partial d$ is assumed to be a continuously differentiable function with respect to $x$, $f$ and $f'$. Although the discriminator $d$ aims to minimize its loss $\mathcal{L}_B$, as mentioned previously, we want the main task model $f$ to challenge the discriminator by providing as little information in $g$ as possible. This essentially requires changing the behavior of $f$ such that the functional $J(f)$ attains its maxima. The extrema (maxima or minima) of functional $J(f)$ can be obtained by finding a function $f$ where the functional derivative of $J(f)$ is equal to zero. By Taylor expansion,

$$J(f) = J(f_{\theta_f^0}) + J'(f_{\theta_f^0})\delta f + o(\delta f) \tag{4}$$

where $\delta f$ is the variation of $f$. Since $o(\delta f)$ is a higher order infinitesimal term at $\theta_f \to \theta_f^0$, and assuming that $o(\delta f)$ is small and ignorable, we can obtain the finite difference approximation of the functional $J(f)$ by plugging it into Eq.(3),

$$\Delta J(f) = J(f) - J(f_{\theta_f^0}) = J'(f_{\theta_f^0})\delta f = \frac{d \int F(x, f, f')dx}{d\theta_f}\delta f \tag{5}$$

If the uniform continuity of $F$ and its derivative is satisfied, we can simplify the derivative of $J(f)$ with respect to $\theta_f$ as follows, while swapping the order of $dx$ with $\delta f$:

$$\Delta J(f) = \frac{\int \partial F(x, f, f')dx}{\partial \theta_f}\delta f = \int \frac{\partial F(x, f, f')}{\partial \theta_f}\delta f dx \tag{6}$$

As $f$ can be any function in an arbitrary function space $\mathcal{F}$. Ensuring the functional derivative of $J(f)$ equal to zero ($\Delta J(f) = 0$) requires $\partial F(x, f, f')/\partial \theta_f = 0$, which is

$$\frac{\partial F(x, f, f')}{\partial \theta_f} = \frac{\partial F}{\partial g} \cdot \frac{\partial g}{\partial f} \cdot \frac{\partial f}{\partial \theta_f} = \frac{\partial F}{\partial g} \cdot \nabla_{\theta_f} loss_A = 0 \tag{7}$$

The last step derivation in above equation is due to our design of letting $g(\cdot)$ be the element-wise loss function of $f$ ($g(\tilde{y}, \hat{y}) = loss_A(\tilde{y}, \hat{y})$).

As an example, suppose we use BCE loss for $\mathcal{L}_B$, and write $p_i$ as the output value of $d(X, \hat{Y}, g(\tilde{Y}, \hat{Y}))$ with index $i$, the Eq.(7) can be written as

$$\frac{\partial F}{\partial \theta_f} = \frac{\partial F}{\partial g} \cdot \nabla_{\theta_f} loss_A = -\sum_{i \in L} \frac{1}{p_i} \nabla_{\theta_f} loss_A(\tilde{y}_i, \hat{y}_i) + \sum_{j \in U} \frac{1}{1 - p_j} \nabla_{\theta_f} loss_A(\tilde{y}_j, \hat{y}_j) \tag{8}$$

Note that the result of above equation can be equivalently perceived as the gradient of a new term $L_f(\tilde{Y}, \hat{Y}, P)$ ($\partial F/\partial \theta_f = -\partial L_f/\partial \theta_f$) with following form:

$$-L_f(\tilde{Y}, \hat{Y}, P) = \sum_{i \in L} \frac{1}{p_i} \cdot loss_A(\tilde{y}_i, \hat{y}_i) - \sum_{j \in U} \frac{1}{1 - p_j} \cdot loss_A(\tilde{y}_j, \hat{y}_j) \tag{9}$$

Consequently, minimizing $-L_f$ with respect to $\theta_f$ will drive the functional derivative of $J(f)$ to zero ($-\partial L_f/\partial \theta_f = \partial F/\partial \theta_f = 0 \implies \Delta J(f) = 0$), which also lead to the maxima of $J(f)$ (update with gradient direction $-\partial F/\partial \theta_f$). We now uncover the exact form of $loss_W$, which can be set as $loss_W(\tilde{Y}, \hat{Y}, P) = -\mathcal{L}_f(\tilde{Y}, \hat{Y}, P)$ to realize the self-interested behavior of $f$. The resulting $loss_W$ can be considered as an additional reweighting loss that complements the original loss term $loss_A$ based on the estimated confidence measures $P$ from the discriminator $d$.

### 3.3 INTERPRETATION OF SCL

Adding $loss_W$ to the original loss term $loss_A$, we can obtain the complete loss for the main task as:

$$\mathcal{L}_A = \sum_{i \in L} \left(1 + \frac{\alpha}{p_i}\right) \cdot loss_A(y_i, \hat{y}_i) + \sum_{j \in U} \left(1 - \frac{\alpha}{1 - p_j}\right) \cdot loss_A(\tilde{y}_j, \hat{y}_j) \tag{10}$$

This essentially transforms the original main task into a cost-sensitive learning problem (Zadrozny et al., 2003) by imposing following soft-labeling weights on the errors of each data label as:

$$\text{Soft-labeling weights} = \begin{cases} 1 + \alpha/p_i, & i \in L \\ 1 - \alpha/(1 - p_j), & j \in U \end{cases} \tag{11}$$

We notice that the above soft-labeling weights induce very different behaviors on the loss $loss_A(\tilde{y}, \hat{y})$ of labeled and pseudo-labeled samples. As $p_i$ represents the judgment of the discriminator $d$ on whether $y_i$ is labeled or not. For labeled samples, the loss are boosted by additional weight of $\alpha/p_i$, which means if the discriminator $d$ judges an labeled samples $y_i \in Y_L$ to be unlabeled or unreliable (confidence $p_i \to 0$), the loss of this sample will get significantly boosted. This forces the main task model $f$ to pay more attention to infer the problematic labeled samples. As unlabeled samples $y_j \in Y_U$ are periodically filled with pseudo-labels generated by $f$ (initially filled with random values), SCL does not fully trust these labels and may even choose to enlarge the gap with the pseudo-label when $p > 1 - \alpha$ ($\alpha \leq 1$). Specifically, the more likely an label $y_j$ is considered reliable by the discriminator ($p_j \to 1$), the stronger it forces the learning process to ignore further improvement on inferring $y_j$.

Note that uniform continuity needs to be satisfied in Eq.(6), however, the derivative of BCE loss will result in $1/p_i$ and $1/(1 - p_i)$ which can potentially take infinite value, violating the continuity assumption. We thus clip their values as $\min(1/p_i, H)$ and $\min(1/(1 - p_i), H)$ in our practical algorithm, where $H$ is a predetermined threshold ($H = 10$ in our implementation). Algorithm 1 presents the practical algorithmic framework of SCL.

---

**Algorithm 1** Self-interested coalitional learning framework

---

**Require:** Training samples $\{X, Y\} = \{X_L \bigcup X_U, Y_L \bigcup Y_U\}$, observability mask $M$, weight parameter $\alpha$
1: Randomly initialized $\theta_f, \theta_d, Y_U$, and let $\tilde{Y}_U \leftarrow Y_U$
2: **while** training **do**
3:      Inference the all data labels $\hat{Y}$ using the main task model $f$
4:      Calculating the element-wise loss of the main task $loss_A(\tilde{Y}, \hat{Y})$ as $g(\tilde{Y}, \hat{Y})$
5:      Predict the label confidence measures $P$ using the discriminator $d$ and $g(\tilde{Y}, \hat{Y})$
6:      Update $\theta_d^{t+1}$ by minimizing $\mathcal{L}_B$ in Eq.(2)      **// Update $d$ with information from main task**
7:      Update $\theta_f^{t+1}$ by minimizing $\mathcal{L}_A$ in Eq.(10)      **// Update $f$ with information from companion task**
8:      update the pseudo-label $\tilde{Y}_U \leftarrow \hat{Y}_U$
9: **end while**

---

## 4 EXPERIMENTS

The proposed SCL framework can be easily incorporated in a variety of SSL tasks. We choose three representative SSL tasks and apply SCL on their official implementations to evaluate the performance of SSL, including: image classification (He et al., 2016) for inductive classification, label propagation (Kipf & Welling, 2017) for transductive classification and data imputation (Gondara & Wang, 2018) for regression.

**Image classification:** We use RestNet18 for image classification on the fashion-mnist dataset (Xiao et al., 2017), with 6,000/10,000 training/testing set partition. Note that the training set only contains 10% of the original fashion-mnist dataset which aims to substantially increase the task difficulty.

**Label propagation:** We trained a two-layer GCN (Kipf & Welling, 2017) integrated with SCL and evaluate the prediction accuracy on Cora dataset(Sen et al., 2008). The test set consists of 1000 labeled examples extracted from a knowledge graph with 2,708 nodes and 5,429 edges. The labeling rate represents the number of labeled nodes used for training divided by the total number of nodes in the training set.

**Data imputation:** We use a deep denoising autoencoder (DAE) (Gondara & Wang, 2018) for data imputation task under regression setting. We use the UCI online news popularity dataset (Asuncion & Newman, 2007) for evaluation. The dataset is split the training/testing set with the ratio of 0.8/0.2.

For all tasks, we remove certain proportion (referred as *missing rate*) of data labels in the training set and treat them as unlabeled data in the semi-supervised setting. At initial step of SCL, as the model is not yet trained to provide pseudo-labels, we fill the initial pseudo-labels for unlabeled data with random labels. For the regression task (data imputation), we compute the mean $\mu$ of the labeled data and fill the missing entries with $\mu$ adding random noise; for the classification tasks, we fill the missing label $Y_U$ by randomly selecting a data label. We present the average results over 10 random seeds for all the experiments. The implementation details and extra comparative experiments between SCL and several advanced SSL algorithms are presented in the Appendix.

| Missing rate (%) | Image classification Accuracy (%) | | Label propagation Accuracy (%) | | Data imputation MSE | |
|---|---|---|---|---|---|---|
| | ResNet18 | ResNet18+SCL | GCN | GCN+SCL | DAE | DAE+SCL |
| 00 | $83.85 \pm 0.12$ | $86.23 \pm 0.13$ | - | - | - | - |
| 10 | $83.57 \pm 0.12$ | $84.69 \pm 0.08$ | $82.18 \pm 1.59$ | $84.33 \pm 0.56$ | 0.8055 | 0.7899 |
| 20 | $83.06 \pm 0.16$ | $85.15 \pm 0.13$ | $82.18 \pm 1.69$ | $84.00 \pm 1.81$ | 0.9029 | 0.8716 |
| 30 | $82.89 \pm 0.21$ | $85.38 \pm 0.16$ | $81.80 \pm 1.75$ | $83.33 \pm 0.68$ | 0.9335 | 0.8867 |
| 40 | $82.43 \pm 0.13$ | $84.98 \pm 0.21$ | $82.28 \pm 2.69$ | $83.00 \pm 0.95$ | 0.9761 | 0.9229 |
| 50 | $81.94 \pm 0.16$ | $84.61 \pm 0.21$ | $81.93 \pm 2.56$ | $81.33 \pm 2.28$ | 0.9726 | 0.9412 |
| 60 | $81.23 \pm 0.16$ | $83.87 \pm 0.20$ | $80.03 \pm 6.37$ | $82.33 \pm 2.10$ | 0.9787 | 0.9727 |
| 70 | $79.70 \pm 0.13$ | $82.90 \pm 0.21$ | $77.84 \pm 7.52$ | $83.67 \pm 3.73$ | 1.1593 | 1.1582 |
| 80 | $78.84 \pm 0.35$ | $81.31 \pm 0.42$ | $74.33 \pm 7.00$ | $80.33 \pm 1.00$ | 1.1429 | 1.1425 |
| 90 | $76.02 \pm 0.41$ | $77.96 \pm 0.47$ | $69.66 \pm 7.66$ | $74.26 \pm 0.53$ | - | |

Table 1: Comparison of the original models and the SCL-integrated version for different tasks (with mean $\pm$ std). The std results in the imputation task are omitted because they are less than 1e-04.

## 4.1 PERFORMANCE EVALUATION OF SCL

We compared our method with the original model in three representative SSL tasks to demonstrate the effectiveness and robustness of SCL under diverse problem setting. Results are summarized in Table 1. We report the classification accuracies and imputation error measured by MSE. SCL consistently achieves superior performance over all tasks, which shows the universality of SCL under diverse SSL problem setting. We also use 10 different levels of missing rates to verify the robustness of SCL. By randomly dropping some labels, we increase the missing rate of the training set from 0% (fully supervised) to 90%. As label propagation and data imputation tasks are not well-defined under fully supervised setting, only the image classification task under 0% missing rates is shown in the table. It is observed in Table 1 that most of the original models experience substantial performance deterioration with the increase of missing rate. By contrast, SCL remains high level of robustness with limited performance drop in most tasks.

Supprisingly, for the inductive image classification task, we observe that in the case of 0% missing rate (fully supervised), SCL can achieve even higher accuracy compared with the fully supervised base model (accuracy: $86.2 \pm 0.13\%$ (SCL) vs $83.85 \pm 0.13\%$ (base model)). Moreover, SCL achieves similar level of accuracy as the fully supervised base model even with $60\%$ missing rate. These results suggests that SCL can also be used in supervised learning problems to boost model performance. SCL performs cost-sensitive learning for different data samples to maximally utilize the hidden information from both labeled and unlabeled data, which lead to more stable and accurate inference under different missing rates. For the transductive classification task like label propagation, all the input data $X$ and the labeled data $Y_L$ are used for inference on the unobserved labels $Y_U$. The amount of information from the labeled data plays a vital role for overall prediction accuracy. Hence as the missing rate becomes large, we observe a substantial performance drop for the original GCN model. SCL, on the other hand, maintains high accuracy in spite of high missing rate by continuously soft pseudo-labeling the data to ensure the reliable pseudo-labels are properly used to facilitate the main task training, thus exhibits high robustness under missing information.

## 4.2 COMPARISON OF SCL AND SELF-TRAINING VIA PSEUDO-LABELING

To further validate the effectiveness of SCL under semi-supervised settings, we compare the performance of SCL with the widely used pseudo-labeling approach in self-training (Yarowsky, 1995; Lee et al., 2013; Laine & Aila, 2017; Iscen et al., 2019b). We compare their performances in three different aspects, i.e., convergence speed, accuracy under different missing rates and the degree of error accumulation under pseudo-labeling. Figure 2 shows us the convergence of image classification, label propagation, and data imputation for missing rates of 40%, 20%, and 30%, respectively. In Figure 2, we can see that the learning speed of the main task model with the corrective reweighting loss term $loss_W$ in SCL is significantly faster than the original model. Meanwhile, since the hard pseudo-labels in self-training are used as true label for model learning rather than carefully reweighted as soft labels in SCL, certain degree of oscillation is observed during the learning process, which impacts the learning speed as well as the final performance of the model. This demonstrates the benefit of using soft-labeling weights to correct unreliable label information in SCL.

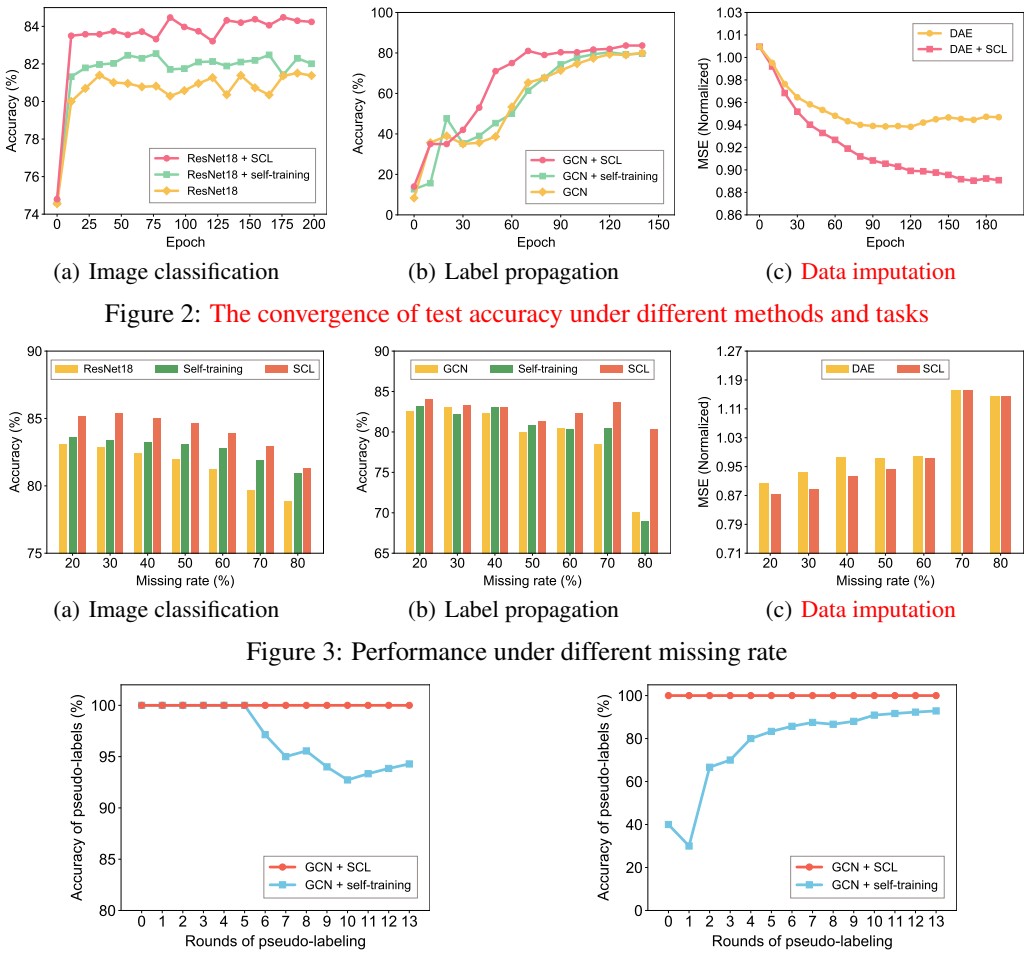

(a) Image classification     (b) Label propagation     (c) Data imputation

Figure 2: The convergence of test accuracy under different methods and tasks

(a) Image classification     (b) Label propagation     (c) Data imputation

Figure 3: Performance under different missing rate

(a) Accuracy of pseudo-labels under 20% missing rate (b) Accuracy of pseudo-labels under 70% missing rate

Figure 4: Error accumulation of different methods in the label propagation task

Figure 3 shows the accuracy of the original, self-trained and SCL versions of models for classification tasks with different missing rates. We observe that SCL maintains high accuracy under different missing rates and consistently outperforms the original and self-training versions. In the label propagation task, although self-training increases the training samples by periodically pseudo-labeling high confidence unlabeled data, the accuracy of the model could decreases due to the introduction of wrong labeling information, which results in subsequent error accumulation and performance drop.

To further investigate the impact of error accumulation in pseudo-labels during model training, we record the accuracy of pseudo-labels after each round of pseudo-labeling in self-training. The test accuracy of these samples in the SCL version of the model are also recorded for comparison. To ensure the stability of self-training, we add pseudo-labels every 10 training epochs. Different from self-training that only label high confidence unlabeled samples, SCL introduces pseudo-labels for all unlabeled data $Y_U$ but with soft-labeling weights. Figure 4 shows the error accumulation trends for self-training and SCL in the label propagation task. When the missing rate is low (Figure 4(a)), self-training maintains a high accuracy at the beginning, but model performance will fluctuate and decrease with the involvement of more potentially problematic pseudo-labels. On the other hand, when the missing rate is high and available information is scarce (Figure 4(b)), gradually involving pseudo-labeled data is beneficial for self-training, resulting in slowly increasing test accuracy, although the model still struggles to achieve high accuracy. Despite the occurrence of error accumulation in self-training, we barely observe such phenomenon in SCL. The test accuracy of the pseudo-labels remains high and stable across different rounds of pseudo-labeling, which clearly suggests the ability of SCL in avoiding error accumulation in pseudo-labels.

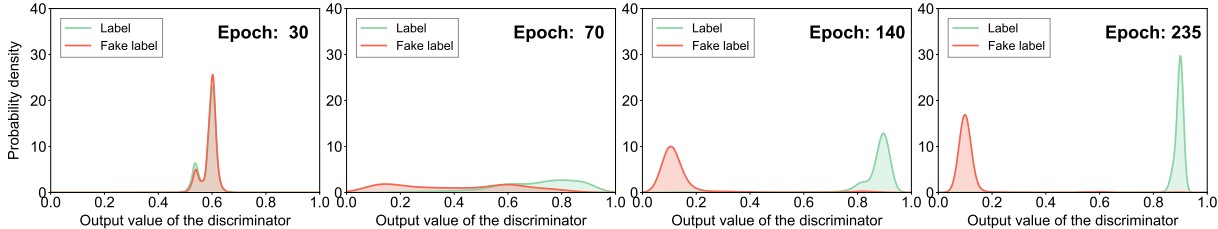

Figure 5: The probability distribution given by the discriminator during the training process

### 4.3 INTERPRETABILITY OF DISCRIMINATOR OUTPUTS

To evaluate the judgement of the discriminator $d$, we analyze its output value distribution on the labeled and intentionally introduced fake/noisy labeled data in different training stages of image classification task. The four plots in Figure 5 clearly show the transition of an improved discriminator, which gradually gains better capability to discriminate true labels ($p \rightarrow 1$) and fake/noisy labels ($p \rightarrow 0$). The interpretability of the discriminator is a byproduct of SCL, which can be particularly useful for many real-world SSL tasks or scenarios involving error or noise in labeled data.

## 5 RELATED WORK

**Self-training via pseudo-labeling.** Pseudo-labeling (Triguero et al., 2015; Yarowsky, 1995; Lee et al., 2013) is one of the most widely used self-training method in semi-supervised learning, which uses a supervised model that is iteratively trained on both labeled and pseudo-labeled data in previous iterations of the algorithm. The basic assumption in most models (Yarowsky, 1995; Lee et al., 2013; Laine & Aila, 2017; Tarvainen & Valpola, 2017b; Iscen et al., 2019b) is that an unlabeled instance can be labeled for training if the underlying model has high confidence in its prediction. When the model makes inaccurate predictions, it is likely to cause error accumulation that negatively impacts training (Cai et al., 2013; Arazo et al., 2020). To alleviate this problem, many studies (Natarajan et al., 2017; Schnabel et al., 2016; Shen et al., 2020; Rizve et al., 2021; **?**) guide the model learning based on various label confidence measures and avoid the model from being overly confident on incorrect pseudo-labels. The above works only focus on leveraging both labeled and unlabeled data, but not fully applicable to scenarios with noisy labeled data. SCL considers the contribution of both labeled and unlabeled data through soft labeling, which better addresses aforementioned drawbacks.

**Learning from noisy labels.** The lack of high-quality labels is common in many real-world scenarios (Natarajan et al., 2017; Schnabel et al., 2016). Erroneous or noisy labels can severely degrade the generalization performance of deep neural networks. Learning from noisy labels (Song et al., 2020) is becoming an increasinly important task in modern deep learning applications. Early studies (Cesa-Bianchi et al., 2011) require unbiased gradient estimates of the loss to provide learning guarantees. Some recent studies (Natarajan et al., 2017; Schnabel et al., 2016; Shen et al., 2020; Algan & Ulusoy, 2021) provide unbiased estimators for evaluating and training under noise data. However, these methods still require that the observed data have a relatively well-behaved (noise-free) underlying distribution, thus the noisy data are weighted according to the same distribution. Satisfying such requirements is difficult in practice and the observed distribution may be biased across the data. In contrast, SCL bypasses the dependence on the original data distribution and learn label confidence measures adaptive using an additional discriminator to facilitate model training.

## 6 CONCLUSION

We introduce a novel self-interested coalitional learning framework, which can be easily incorporated into a wide range of semi-supervised learning algorithms for enhanced performance. SCL improves semi-supervised model learning by constructing a semi-cooperative "game" that establishes cooperation between the main self-interested semi-supervised learning task and a companion task of inferring the observability of labels. Meanwhile, it also offers additional confidence measures for the original and inferred data labels, which can be particularly useful for the practical applications with noisy labels. Through comprehensive evaluations, we show that SCL can consistently improve the performance and robustness of semi-supervised classification and regression algorithms.

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

## A    EXPERIMENT SETTINGS

- **Image classification.** The base learner ($f$) is a ResNet18 (He et al., 2016) where the input channels of the first convolutional layer are replaced with 1. We use the Kaiming Normalization (He et al., 2015) to initialize the parameters of $f$. All input samples of training set and test set are transformed with a mean and a standard deviation for normalization. The discriminator consists of two feature extractors for $X$ and $hatY$, respectively, and a fusion layer whose inputs are the extracted $X$ and $hatY$ features and the main task information $g$. The feature extractor of $X$ is also a ResNet18 with the number of input channels of the first convolutional layer as 1, and it outputs features with dimension 1. The feature extractor for $hatY$ is a network with three-layer FC, where the first two layers are each followed by a ReLU layer. We fuse the extracted features of $X$ and $hatY$ by the Hadamard product. And concatenate the loss $g$ to the fused embeddings of $X$, $hatY$ and fuse them through a one-layer FC. We feed the output into a sigmoid layer to output the probabilities. We use the Adam optimizer with a learning rate of 0.001 to train the base learner $f$ and discriminator $d$ simultaneously for a total of 300 epochs. We set $\alpha = 0.6$ in all experiments for this task.

- **Label propagation.** We train a two-layer GCN as described in (Kipf & Welling, 2017) and evaluate prediction accuracy on a test set of 1,000 labeled examples. We initialize weights using the initialization described in (Glorot & Bengio, 2010) and accordingly (row-)normalize input feature vectors. For discriminator $d$, we use a five-layer and two-layer FC to encode the input $X$ and $\hat{Y}$, and we fuse the hidden embedding of $X$ and $\hat{Y}$ by hadamard product. We connect $g$ to the fusion embedding of $X, \hat{Y}$ and fuse it through one-layer FC and then fed to a sigmoid layer to give the final probability. All FC layers is followed by a ReLU operations except the output layer. We train all models for a maximum of 150 epochs (training iterations) using Adam (Kingma & Ba, 2015) with a learning rate of 0.01. We set $\alpha = 1$ in all experiments for this task.

- **Data imputation.** The denoise autoencoder we used in the data imputation task as described in (Gondara & Wang, 2018) consists of a dropout layer and an encoder composed of four-layer FC and a decoder composed of four-layer FC, where the dropout layer has $p = 0.1$. For discriminator $d$, we use three two-layer FC to encode the input $X, \hat{Y}$ and $g$, and we fuse the hidden embeddings by two one-layer FC and then fed to a sigmoid layer to give the final probability. All FC layers in the discriminator are followed by ReLU operations, while all FC layers in the DAE are followed by tanh operations, with the exception of the output layer. We train all models for a maximum of 100 epochs using Adam (Kingma & Ba, 2015) with a learning rate of $10^{-5}$ and early stopping with a window size of 10, i.e. we stop training if the validation loss does not decrease for 10 consecutive epochs. We set $\alpha = 1$ in all experiments for this task.

## B    ADDITIONAL EXPERIMENT RESULTS

### B.1    ADDITIONAL ABLATION STUDY

In this section we present the ablation study of the weight parameter $\alpha$ under the image classification task to analyze the impact of $\alpha$ in SCL. We selected three different levels of missing rates in our experiments - low (30%), medium (50%) and high (80%). Figure 6 shows the classification accuracy obtained by SCL using different $\alpha$ values under different missing rates. We observe that a larger $\alpha$ will make the main task favors more on minimizing the reweighting loss rather than simply minimizing the difference between model predicted labels and the given labels $\tilde{Y}$. In our exeriment results in Figure 6, it actually suggests that with the $\alpha$ taking value from the recommended range $[0.1, 0.9]$, SCL is not very sensitive to $\alpha$. The maximum difference of accuracy between the worst and best choice of $\alpha$ is only about 2% under different missing rates. The insensitivity to $\alpha$ is another advantage of SCL, as we do not need to perform complex parameter tuning in order to learn a robust and high performance model, which is very implementation friendly.

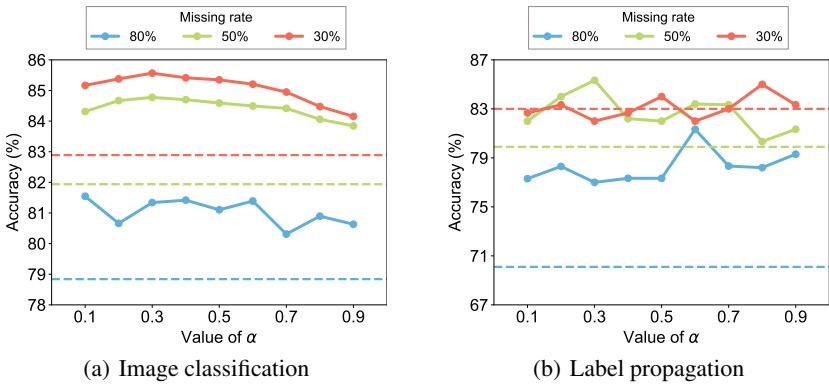

(a) Image classification        (b) Label propagation

Figure 6: Test accuracy under different weight parameter $\alpha$

## B.2 COMPARISON WITH ADVANCED SSL METHODS

We conduct supplementary experiments on the semi-supervised image classification task on CIFAR-10 dataset with 4000 labeled data samples. We compare SCL with several advanced semi-supervised learning methods, including the state-of-the-art method: uncertainty-aware pseudo-label selection (UPS) (Rizve et al., 2021). We use the CNN-13 architecture that is commonly used in benchmarking SSL algorithms (Oliver et al., 2018) and set $\alpha = 0.9$ for SCL in this task. For a fair comparison, we compare against the methods which report results using the same CNN-13 backbone architecture: DeepLP (Iscen et al., 2019a), TSSDL (Shi et al., 2018), MT (Tarvainen & Valpola, 2017a), MT + DeepLP, ICT (Verma et al., 2019), DualStudent (Ke et al., 2019), and MixMatch (Berthelot et al., 2019). We keep the experiment setting the same as in the UPS paper (Rizve et al., 2021) for comparison. The technical details of above baselines algorithms please refer to their papers.

We present in Table 2 the mean and standard deviation of each method across three different random splits on training and testing data. SCL achieves comparable results to the state-of-the-art method UPS (92.2% accuracy as compared to 93.6% in UPS), despite that SCL is much more light-weighted and simpler as compared to UPS. Under the same hardware setting (8-core 4.7GHz CPU and Nvidia RTX3070 GPU), it takes 19 hours to train UPS to reach 92% accuracy and at least 60 hours to fully converge; while SCL takes only 7 hours to reach final convergence, a 60% reduction in training time to reach the same level of accuracy. UPS employs the uncertainty measure to select high-quality pseudo-labels and also uses negative sample learning to improve the model performance. The uncertainty-aware pseudo-label selection scheme used in UPS is very costly, which lead to high training cost. By contrast, in SCL, all unlabeled data are pseudo-labelled and used for training, but with a soft-labeling weights to unevenly update the loss of different samples. It does not involve any pseudo-label selection process as used in most existing pseudo-labeling methods. Moreover, SCL is a very general SSL framework which broadly applicable to semi-supervised inductive classification, transductive classification and regression tasks, whereas most of the previous baseline algorithms are only applicable to semi-supervised inductive classification tasks. This shows the great potential and computational efficiency of using SCL solving real-world large-scale applications.

Table 2: Error rate (%) on the CIFAR-10 test set. Methods with * are pseudo-labeling based.

| Methods | CIFAR-10 (4000 labels) |
|---|---|
| DeepLP* | $12.69 \pm 0.29$ |
| TSSDL* | $10.90 \pm 0.23$ |
| MT | $11.41 \pm 0.25$ |
| MT+DeepLp | $10.61 \pm 0.28$ |
| ICT | $7.29 \pm 0.02$ |
| DualStudent | $8.89 \pm 0.09$ |
| MixMatch | $6.48$ |
| UPS* | $6.39 \pm 0.03$ |
| SCL | $\mathbf{7.85 \pm 0.03}$ |

