# OpenReview forum: "Enhancing semi-supervised learning via self-interested coalitional learning"
_ICLR.cc/2022/Conference — ICLR 2022 Submitted_

### Official Review · Reviewer_HSTc · 2021-10-27

**Correctness:** 2
**Technical Novelty And Significance:** 2
**Empirical Novelty And Significance:** 2
**Recommendation:** 3
**Confidence:** 4

**Main Review:**

The idea of introducing the auxiliary task that discriminates whether an instance is labeled is quite interesting. In effect, such a strategy is first introduced in active learning [1]. In the VAAL method [1], a similar discriminator is introduced to identify where an example is labeled or not, which is then used to indicate the uncertainty of an example for active selection. Therefore, the proposed method has a close connection to a recent work in SSL [2] that also employs the uncertainty measure to select high-quality pseudo-labels.

I have the following concerns.

1. The derivation in section 3.2 is confusing. For example, in Eq. (3), the second equality is incorrect and the term $\frac{dd}{dx}$ should be added. Also, it would be better to change the notation of $d$ (discriminator) to another one, since derivative dx also uses the notation d. Besides, I actually did not understand why $\mathcal{L}_B$ depends on $f$, since $f$ and $\mathcal{L}_B$ are from two different branches without sharing network blocks (Figure 1).
2. The experimental section is not convincing and this is my main concern. The datasets and the baselines are too simple. State-of-the-art SSL methods should be employed to support the claims. In particular, the uncertainty-based SSL method [2] should be compared. As I have discussed above, the proposed method can implicitly be equal to existing techniques in SSL. Is the proposed method complementary to existing methods? Or it is contradictory to some techniques? These questions require an in-depth empirical analysis.

Overall, this work is below the bar of an ICLR paper regarding its poor experiments.


[1] Sinha S, Ebrahimi S, Darrell T. Variational adversarial active learning[C]//Proceedings of the IEEE/CVF International Conference on Computer Vision. 2019: 5972-5981.

[2] Rizve M N, Duarte K, Rawat Y S, et al. In defense of pseudo-labeling: An uncertainty-aware pseudo-label selection framework for semi-supervised learning[J]. arXiv preprint arXiv:2101.06329, 2021.

**Summary Of The Paper:**

This paper proposes a new semi-supervised learning framework by introducing an auxiliary task that distinguishes whether the pseudo-labels are truly labeled or not. Then, this information is used to add a reweighting loss to the main objective. Experiments on several simple benchmark datasets show that the proposed method outperforms some naive baselines.

**Summary Of The Review:**

Interesting idea, poor experiments and confusing derivation.

---

> ### Author Response · Authors · 2021-11-21
> **Response to Reviewer HSTc**
>
> **"The derivation in section 3.2 is confusing. For example, in Eq. (3), the second equality is incorrect and the term dddx should be added . Also, it would be better to change the notation of d (discriminator) to another one, since derivative dx also uses the notation d. Besides, I actually did not understand why LB depends on f, since f and LB are from two different branches without sharing network blocks  (Figure 1)."**
>
> **RESPONSE:** The derivation of SCL is based on functional analysis and calculus of variation, it might cause some confusion if the readers are not familiar with this area. To clarify the confusion of the reviewer, we provide the detailed reasoning behind our derivation as follows:
> * The overall derivation logic of SCL is as follows. As we introduce information $g(\cdot)$ in the input of the discriminator $d$, where $g(\cdot)$ has the form of element-wise loss term $loss_A$ of $f$, which is depend on $f$, hence $d$ as well as its loss $\mathcal{L}_B$ now become the functionals of $f$ (i.e. function of a function). More specifically, $\mathcal{L}_B$ can be considered as $\mathcal{L}_B(\theta_d, f)$. To solely analyze the impact of $f$ on $\mathcal{L}_B$, we instead define a new functional $J(f)$ for $\mathcal{L}_B$ which has the parameter of the discriminator $\theta_d$ fixed and no longer considered as a variable. We maximize $J(f)$ with respect to $f$ in order to enforce the self-interested behavior of $f$, and use calculus of variation to identify a necessary condition for the maxima of $J(f)$ ($\partial F/\partial \theta_f=0$). This can then be perceived as minimizing an additional reweighting loss term $\mathcal{L}_w$ for the main task model $f$, and eventually transform the original main task into a cost-sensitive learning problem.
> * A key step in our derivation is to introduce the new functional $J(f)$ for  $\mathcal{L}_B$, and analyze upon $J(f)$ instead of $\mathcal{L}_B$ to allow decoupled analysis of $f$ and $d$. As $J(f)$ is a functional of $f$ and $f$ depends on $x$, following a commonly used trick in calculus of variation, we write functional $J(f)$ in an integral form of a functional $F(x, f, f^\prime)$ (has the form of $\partial \mathcal{L}_B/\partial d$ but with $d$ fixed) to eliminate the effect of changes in $x$ when dealing with the variation of $f$. Such a treatment is also used in deriving the well-known Euler-Lagrangian equation. Note that as $d$ is fixed and not a variable for $J(f)$, it does not have any gradient upon $J(f)$ or $F(x, f, f^\prime)$, hence the gradient of $d$ ($\text{d}d$) should not appear in Eq.(3).
> * Furthermore, there is no gradient passing between the main task model $f$ and discriminator $d$ in our model. Both the main task model $f$ and the discriminator $d$ are learned in a fully supervised manner with their own optimization objective (see Eq.(2))) respectively. The information $g(\cdot)$ from $f$ and the discriminator output values $d(x,y ,g(\cdot))$ used in the additional loss term $\mathcal{L}_w$ are passed as values.
>
> **"The experimental section is not convincing and this is my main concern. The datasets and the baselines are too simple. State-of-the-art SSL methods should be employed to support the claims. In particular, the uncertainty-based SSL method [2] should be compared.  As I have discussed above, the proposed method can implicitly be equal to existing techniques in SSL. Is the proposed method complementary to existing methods? Or it is contradictory to some techniques? These questions require an in-depth empirical analysis."**
>
> **RESPONSE:** We thank the reviewer for this comment. As described in the general response, we have added more supplementary experiments and compared SCL with several advanced pseudo-labeling methods, including the state-of-the-art method UPS as suggested by the reviewer. We have revised the experiment section as well as the Appendix to report these results. SCL offers very competitive performance while has much higher computational efficiency compared with SOTA SSL methods.
>
> Also, as we have discussed in the previous responses, SCL is very different from existing SSL methods. Most of the existing pseudo-labeling methods (e.g. Ref [2] mentioned by the reviewer, Ref [3]) are based on reliably selecting pseudo-labels to alleviate the error accumulation problem. There are also studies (e.g. Ref [4],[5]) that adjust weights on the pseudo-labeled data according to their data distribution by some confidence measures, which mostly rely on the observable data distribution. While in SCL, all unlabeled data are pseudo-labelled and used for training, but with a soft-labeling weights to unevenly update the loss of different samples. It does not involve any peudo-label selection process nor depends on the original data distribution, but rather, we use the output of the jointly learned discriminator to construct a reweighting loss term $loss_W$ for the main task model $f$.

---

> > ### Author Response · Authors · 2021-11-21
> > **References for response**
> >
> > [3] Yan Zhou, Murat Kantarcioglu, Bhavani M. Thuraisingham. Self-Training with Selection-by-Rejection. ICDM 2012: 795-803
> >
> > [4] Kumar, Ananya, Tengyu Ma, and Percy Liang. “Understanding self-training for gradual domain adaptation.” International Conference on Machine Learning. PMLR, 2020.
> >
> > [5] Wenzheng Feng, Jie Zhang, Yuxiao Dong, Yu Han, Huanbo Luan, Qian Xu, Qiang Yang, Evgeny Kharlamov, Jie Tang. Graph Random Neural Networks for Semi-Supervised Learning on Graphs. NeurIPS 2020

---

> ### Author Response · Authors · 2021-11-21
> **Response to Reviewer HSTc**
>
> **"The idea of introducing the auxiliary task that discriminates whether an instance is labeled is quite interesting. In effect, such a strategy is first introduced in active learning [1]. In the VAAL method [1], a similar discriminator is introduced to identify where an example is labeled or not, which is then used to indicate the uncertainty of an example for active selectio n. Therefore, the proposed method has a close connection to a recent work in SSL [2] that also employs the uncertainty measure to select high-quality pseudo-labels."**
>
> **RESPONSE:** Introducing a discriminator is only one insight of SCL. The more important contribution in our paper is to introduce a cooperative-yet-competitive learning framework to enable information sharing between the two tasks, while at the same time allowing the decoupled training of both tasks with their own learning objectives in a fully supervised manner (see Eq.(2)), which is very easy to train and computationally cheap.
> Another unique characteristic of SCL compared with previous pseudo-labeling approaches, is that for every training step, all unlabeled data are pseudo-labelled and used for training, but with a soft-labeling weights to unevenly update the loss of different samples. This is very different from most existing pseudo-labeling methods, which primarily focus on select high-quality pseudo-labels and only use the selected pseudo-labels for training.
> Regarding the detailed difference of SCL and the mentioned two references, we provide detailed discussion as follows:
>
> * The similarity of SCL and VAAL [1] is only that both methods use a discriminator for learning. However in VAAL, there is no information sharing or the self-interested behavior as we introduce in SCL. VAAL simply adopts an GAN framework which trains a VAE as generator and use the discriminator for discriminating. However, as we have discussed in the paper, the scheme used in SCL is very different from typical GAN-style framework, which establishes a cooperative-yet-competitive "game" between the two tasks. Our approach allows much easier and computationally efficient training, whereas the GAN-style framework requires to solve the complex and costly min-max optimization problem and often suffers from issues such as traning instability and mode collapse.
> * UPS [2] employs the uncertainty measure to select high-quality pseudo-labels and also uses negative sample learning to improve the model performance. Although UPS performs strongly in benchmark tests, its uncertainty-aware pseudo-label selection scheme is very costly. We have added the comparative experiment between UPS and our method. Under the same hardware setting (8-core 4.7GHz CPU and Nvidia RTX3070 GPU), it takes 19 hours to train UPS to reach 92\% accuracy and at least 60 hours to fully converge; while SCL takes only 7 hours to reach final convergence, a 60\% reduction in training time to reach the same level of accuracy, and also achieves competitive performance. Please refer to our revised paper for detailed results.

---

> > ### Comment · Reviewer_HSTc · 2021-11-22
> > **Further discussion**
> >
> > Thanks for the detailed response.
> >
> > #### 1. About my former comments
> > My concern on the derivation has been addressed. However, in terms of the experimental results, I remain unsatisfactory. Though additional empirical results slightly alleviated my concern, there is still room to explore the effectiveness of SCL. For example, as Reviewer 3FG6 has requested, incorporating SCL with the recent prevalent local consistency technique would be particularly interesting. Moreover, only results on CIFAR-10 were reported. What about more challenging tasks like CIFAR-100 and ImageNet? If this paper is accepted, it is unfair for other papers which provide comprehensive empirical results alongside the original submission.
> >
> > #### 2. Need further clarification in terms of a relevant work
> > I have noted that the **self-interested coalitional learning technique is originally proposed in a KDD 2021 paper [1], where the proposed method is also called SCL**. The proposed formulation in [1] is quite similar to this ICLR 2022 submission. For example, Equation (1), (4), (7-9) in [1] are similar to the Equation (1), (2), (9-11). However, in the manuscript, the authors did not well clarify the difference between these two works but used [1] to support some claims like 'the output of the discriminator is in fact a measure of label reliability, thus can be interpreted as the confidence probabilities of labels' and 'jointly minimizing the augmented loss may impede both tasks from achieving the best performance'.
> >
> > The biggest difference between this work and [1] is that [1] addresses the **Network-Wide Traffic States Imputation**, where part of input $x$ is given and the goal is to complete the missing inputs. In that regard, semi-supervised learning is also similar, where the goal is to complete the missing labels. It might be the reason that the authors used the data imputation task to support the effectiveness of the proposed method.
> >
> > So, did the authors intentionally overlook the relevance?
> >
> > I acknowledge that these two works address different tasks, but I believe the novelty would be limited since most of the techniques of this submission followed [1].
> >
> > [1] Qin H, Zhan X, Li Y, et al. Network-Wide Traffic States Imputation Using Self-interested Coalitional Learning[J]. KDD 2021. https://dl.acm.org/doi/10.1145/3447548.3467424.

---

> > > ### Author Response · Authors · 2021-11-23
> > > **Further discussion for Reviewer HSTc**
> > >
> > > **Regarding adding experiments on larger datasets:**
> > >
> > > We thank the reviewer for the suggestion. However, limited by the relatively short author response period, we are only able to replicate the suggested new baseline methods and finish the additional comparative experiments on Cifer-10 dataset. Running all experiments on larger datasets like ImageNet requires substantially longer time under our limited computation resources. As we have reported in the paper, training UPS on even Cifer-10 dataset takes at least 60 hours to converge.
> > >
> > > Moreover, for many SSL problems, the key challenge is the lack of labeled data rather than the size of the data. In our paper, we conducted our evaluation on a wide variety of SSL tasks (e.g. inductive and transductive classification, data imputation) as well as additional comparative study with advanced SSL algorithms. SSL consistently achieves strong performance and very high computational efficiency. We do not find any other SSL study has such a broad coverage nor offer the same level of generalizability as in our paper.
> > >
> > > **Difference with the KDD 2021 paper:**
> > >
> > > Our paper has several key differences with the KDD 2021 paper [1]:
> > >
> > > 1. We propose a general learning framework that is applicable to a wide-range of SSL tasks, while paper [1] mainly focuses on missing data imputation problem, with the key emphasize on the spatio-temporal modeling of the data.
> > > 2. Paper [1] has some theoretical gap when deriving the final reweighting factors. For example, the treatment of setting the derivative of the discriminator equal to 0 is given in terms of intuition rather than solid theoretical analysis. In our paper, we provide concrete theoretical derivation of a similar learning process using functional analysis and calculus of variation. We showed that it is actually not setting the derivative of the discriminator equal to 0 that works, but rather, we need to find the maxima of the functional $J(f)$ of $\mathcal{L}_B$， which happens embody the self-interested behavior of the main task model $f$.
> > > 3. The method proposed in paper [1] does not involve the self-training process as in our framework, which can not be applied to general SSL setting. In paper [1], all missing data are filled with the random values, and the final learning objective minimizes the error on observed values and enlarges error on random filled data (e.g. with negative error weights). This is not suitable for general SSL tasks, as simply enlarge the error on randomly filled unlabeled data forbids the effective use of extra information in unlabeled data. Our framework is formulated upon the pseudo-labeling paradigm, which iteratively updates the pseudolabel of unobservable samples, and use discriminator outputs to reweight different pseudo-labeled data during training. As a result, the final soft-labeling weights developed in our framework is different from paper [1]. Specifically, our method allows adaptively minimizing or enlarging the gap with the pseudo-labeled data depending on their reliability measured by the discriminator, e.g. whether $1-\alpha/(1- p)>0$ or not (see Eq.(11) and the subsequent discussion of our paper).

---

> > > > ### Comment · Reviewer_HSTc · 2021-11-23
> > > > **Response**
> > > >
> > > > Thanks again for your detailed response. I did not ask you for more experiments on large-scale datasets and the difference between the two works.
> > > >
> > > > ### About experiments
> > > >
> > > > While the authors blame that There is a lack of time to conduct more experiments, I have to say that this should have been done when you submitted this manuscript. **As I said, it is not fair for other researchers who provided comprehensive experiments alongside their submission.**
> > > >
> > > > ### About the difference
> > > >
> > > > I did not ask for the differences since I have read these two papers. Since most techniques of this submission followed the KDD 2021 paper, the authors should clarify the differences at the beginning.
> > > >
> > > > My question is: **Why didn't you clearly indicate the difference between these two papers in the original submission?**
> > > >
> > > > Besides, I have new concerns about the novelty.

---

### Official Review · Reviewer_dWui · 2021-10-31

**Correctness:** 4
**Technical Novelty And Significance:** 3
**Empirical Novelty And Significance:** 2
**Recommendation:** 6
**Confidence:** 3

**Main Review:**

Pros:
- Overall the paper is well-structured and easy to follow.
- The new method achieves its original goals and improves SSL effectiveness by jointly solving the main and the auxiliary tasks.
- The authors introduce a new SCL strategy to solve the problems, which can be applied to a broader class of learning problems.

Cons:
- Lack of experiments
    - The proposed method is only compared with the self-learning method (with the same base learner). While this demonstrates how the model is improved with SCL, it is also necessary to compare with state-of-art SSL methods.
    - It's also valuable to include the supervised method with fully-labeled dataset as a reference in all experiments.
    - For data imputation, a more common case is that missing state is correlated with input/output instead of simply random missing. It also checks the method robustness against labeled/unlabeled distribution shift.
    - Compared with original self-learning method, the new method has an extra discriminator model, which are based on the same base learners as for the main tasks. It's meaningful and more fair to compare with supervised models of higher capacity.
- The paper doesn't cover how SCL can work together with consistency regularization, which is commonly used together with self-learning.


Besides, I have a few questions:
- Although Table 1 doesn't have a row for Missing rate = 0% (full dataset), it seems SCL methods have better accuracy than model trained with full dataset for the first two tasks. Is this because the SCL has double model capacity due to the extra discriminator?
- Why is the test accuracy of pseudo-labels 100% for SCL method in Figure 4? Are they calculated differently?

**Summary Of The Paper:**

This paper proposes a new semi-supervised learning method. Motivated by the error accumulation problem of typical self-training paradigms, the authors propose to explicitly model the confidence of pseudo labels as an auxiliary task. They come up with a self-interested coalitional learning (SCL) strategy to solve both tasks jointly. Under the new framework, the main task is transformed into a cost-sensitive learning problem. Experiments demonstrate that pseudo labels are substantially more accurate with the new method and better performance of the main tasks at different label missing rates.

**Summary Of The Review:**

This is an interesting paper from technique perspective. But it definitely needs more empirical studies to demonstrate practical value.

---

> ### Author Response · Authors · 2021-11-21
> **Response to Reviewer dWui**
>
> We thank the reviewers for the comment. Regarding the concerns from the reviewer, we describe the discussion as follows:
>
> **"The proposed method is only compared with the self-learning method (with the same base learner). ... it is also necessary to compare with state-of-art SSL methods."**
>
> **RESPONSE:** We have added supplementary experiments on CIFAR-10 dataset and compared SCL with several advanced pseudo-labeling methods, including the state-of-the-art method: uncertainty-aware pseudo-label selection (UPS) (Rizve et al. ICLR 2021). We have revised the experiment section as well as the Appendix to report these results. SCL offers very competitive performance while has much higher computational efficiency compared with SOTA SSL methods.
>
> **"It's also valuable to include the supervised method with fully-labeled dataset as a reference in all experiments."**
>
> **RESPONSE:** We thank the reviewer for this comment. We have included additional experiment for the image classification task with $0\%$ missing rate to compare with the fully supervised setting (see revised Tabel 1). SCL achieves higher accuracy compared with the fully supervised base model, which demonstrate the possible applicability of SCL in areas beyond semi-supervised learning.
> Please refer to our general response for detailed discussion and analysis.
>
> **"For data imputation, a more common case is that missing state is correlated with input/output instead of simply random missing.  It also checks the method robustness against labeled/unlabeled distribution shift."**
>
> **RESPONSE:** Our main purpose in this paper is to propose a general framework that are broadly applicable to many SSL tasks. The data imputation problem with structured missing pattern that depend on missing states is interesting, but it is beyond the scope of this paper. To adequately solve this problem, one might need to introduce some prior knowledge about the dependency structure of the data missing pattern, which requires special design for the main task model. Simply relying on a semi-supervised learning algorithm may not be sufficient to best address this problem.
>
> **"Compared with original self-learning method, the new method has an extra discriminator model, ... It's meaningful and more fair to compare with supervised models of higher capacity. "**
>
> **RESPONSE:** We agree with the reviewer that introducing the extra discriminator in some sense increases the model capacity for our main task model. However, the main purpose of our paper is to propose a general purpose semi-supervised learning framework rather than improving performance for supervised learning models. Hence in our revised paper, we focuse on adding supplementary comparative experiments to evaluate SCL against the state-of-the-art SSL methods. Please find our revised Experiments Section and Appendix for detailed resulsts.
>
> **"The paper doesn't cover how SCL can work together with consistency regularization, which is commonly used together with self-learning."**
>
> **RESPONSE:** In theory, SCL does not conflict with consistency regularization. As the main task model in SCL is trained on all data samples, with additional reweighting loss posed on samples. It is possible to use both types of SSL method together to further improve the learning ability of the model as long as the consistency regularization loss can be explictly formulated and considered as part of the main task loss $loss_A$.
> But our goal is to develop a general SSL framework, applicable to the widest range of SSL problems, hence techniques with domain-restrictions such as consistency regularization are not evaluated in this paper.
>
> **"Although Table 1 doesn't have a row for Missing rate = 0% (full dataset), ... Is this because the SCL has double model capacity due to the extra discriminator? "**
>
> **RESPONSE:** We have added the results for the 0\% missing rate case in Table 1 of the revised paper. It is true that SCL can achieve better accuracy than the fully supervised model. We provide detailed discussion and our conjecture on this phenomenon in the general response.
>
> **"Why is the test accuracy of pseudo-labels 100% for SCL method in Figure 4?  Are they calculated differently?"**
>
> **RESPONSE:** In Figure 4, we record the samples selected by the self-training method in each round of pseudo-labeling, and calculate their label prediction accuracy under SCL. Note that self-training method only select the high-confidence samples for pseudo-labeling (in our experiment, only select samples with class prediction probability greater than 0.99). Hence these selected samples are themselves easier to learn with the main task model compared with the rest of unlabeled samples. Also, as we have shown in Figure 2, SCL has a faster learning curve compared with self-training method, therefore it is very possible that main task model in SCL may have higher accuracy on these selected easily learnable pseudo-labeled samples.

---

### Official Review · Reviewer_Qifr · 2021-10-31

**Correctness:** 4
**Technical Novelty And Significance:** 4
**Empirical Novelty And Significance:** 4
**Recommendation:** 5
**Confidence:** 3

**Main Review:**

This is an interesting paper on a topic with important practical applications: semi-supervised learning. The contribution appears to be original, and it is likely to influence future work in the field. The authors are explicitly calling out and addressing the two main weaknesses of traditional self-learning approaches: error accumulation and over-reliance on the labeled data.

The paper would greatly benefit from an additional section that would provide an intuitive, illustrative example of how and why the proposed approach outperforms self-training. Ideally, it should compare and contrast the convergence of (1) self training, (2) the auxiliary task, and (3) SCL.

The paper would also benefit by tightening the narrative around the ALPHA parameter, which, in the main paper, is only discussed in the theoretical framework. Appendix A provides no value of ALPHA for the first dataset, and it proposed (without any justification) a value of 1 for the other two domains. Appendix B is extremely brief and not very helpful. The authors make no recommendation on how to tune alpha, and the argument that even the worst alpha (in the 0.1 - 0.9 range) is better than the original model is fairly weak, given the wide variations of the accuracy due to changes the value of alpha.

OTHER COMMENTS:
- for Table 1, please add three more rows: 0%, 90%, and 99%.  The former is critical to understanding the upper-bound performance, while the later two will bring SCL into a more realistic semi-supervised regime, where unlabeled data is one or two orders of magnitude more abundant than the labeled data
- please add to Figure 6 the horizontal lines with the accuracy of the original model for each of the three missing rates
- it is still unclear why did you choose to use only 10%of the data for image classification (page 6); is scalability to large datasets a concern?
- please spell-check the paper - eg, "perforamnce" on page 4

- page 2:  please replace "more sufficient"
- page 3: "jointly solving above two tasks"   -->  "jointly solving THE above two tasks"
- page 3: "there are some other works embody" --> "there are some other works THAT embody"
- page 4: "are impacted the influence" -->  "are impacted BY the influence"
- page 7: please replace "well learn"

**Summary Of The Paper:**

The paper introduces Self-interested Coalitional Learning (SCL), which is a novel approach to semi-supervised learning. SCL combines the traditional self-training approach to semi-supervised learning with an auxiliary task that infers label observability. The empirical results show that, in a variety of scenarios, SCL outperforms both self-training and the original model.

**Summary Of The Review:**

Overall, this paper uses a novel idea to improve the state of the art for semi-supervised training.

---

> ### Author Response · Authors · 2021-11-21
> **Response to Reviewer Qifr**
>
> **"For Table 1, please add three more rows: 0%, 90%, and 99% . The former is critical to understanding the upper-bound performance, while the later two will bring SCL into a more realistic semi-supervised regime, where unlabeled data is one or two orders of magnitude more abundant than the labeled data."**
>
> **RESPONSE:** We appreciate the reviewer for the constructive suggestion. We have conducted additional experiments to address the reviewer's concern. Since label propagation and data imputation tasks are not well-defined under fully supervised setting, we have added experiments under 0%, 90% missing rates for the image classification task, and these results are added in Table 1. We did not include the results for $99\%$ missing rate, as there are only 6 labeled samples in the training set of the classification task, and all methods including the baselines failed to give meaningful results.
>
> As discussed in the general response, in the case of 0% missing rate (fully supervised), SCL achieves higher accuracy compared with the fully supervised base model (accuracy: $86.2\pm 0.13\%$ (SCL) vs $83.85\pm 0.13\%$ (base model)). Detailed discussion and analysis please check our general response.
>
> **"Please add to Figure 6 the horizontal lines with the accuracy of the original model for each of the three missing rates."**
>
> **RESPONSE:** We thank the reviwer for the suggestion. We have added it in this revision. Please check the our revised paper for details.
>
> **"It is still unclear why did you choose to use only 10% of the data for image classification (page 6); is scalability to large datasets a concern? "**
>
> **RESPONSE:** Since image classification on the fashion-mnist dataset is not a very hard task given image classification models such as ResNet18. Most models can achieve very high accuracy without the need of introducing any semi-supervised algorithms if missing rate is not very high. To make the task more challenging, we reduce the amount of data for model learning and use only 10% of the dataset to to test the model performance.
>
> **"Typos in the paper."**
>
> **RESPONSE:** We thank the reviewers for pointing out these typos and providing constructive suggestions for improvement. We have corrected them in our revised paper.

---

> ### Author Response · Authors · 2021-11-21
> **Response to Reviewer Qifr**
>
> We thank the reviewers for the positive feedback and recognition of the originality of this work.
> We provide detailed response regarding the reviewer’s questions as follows.
>
> **"The paper would greatly benefit from an additional section that would provide an intuitive, illustrative example of how and why the proposed approach outperforms self-training.  Ideally, it should compare and contrast the convergence of (1) self training, (2) the auxiliary task, and (3) SCL."**
>
> **RESPONSE:** Although SCL also incorporates the self-training strategy, it operates very differently compared with most existing self-training methods. In SCL, all unobserved labels are pseudo-labeled, and SCL uses the confidence measures provided by the discriminator to learn from all samples but with different weights. This is very different from most existing pseudo-labeling based self-training methods, which focus primarily on select high-quality pseudo-labels and only use the selected pseudo-labels for training. The scheme used in SCL allows it leverage information from all data for training, hence allieviate the issue of over-reliance on the labeled data and can often achieve accelerated learning speed as shown in Figure 2.
>
> We have actually discussed and compared the performance of SCL against the vanilla model (no auxiliary task) and self-training in Section 4.2. Taking the image classification task as an example, Figures 2 present the learning curves of origianl model, models with self-training and SCL; and Figure 5 presents the discriminating results of fake labels at different training stages of the auxiliary task, which can be used to inspect the learning progress of the discrimator. From these two figures, we find SCL learns faster compared with the vanilla model and the model with self-training. The main task needs about 200 epochs to converge, while for the auxiliary task, at about 140 epochs, the discriminator can already provide reasonable probability outputs. Despite that the discriminator is a light-weighted model in SCL, it offers substantial improvement for the main task without adding too much computation burden.
>
> **"The paper would also benefit by tightening the narrative around the ALPHA parameter, which, in the main paper, is only discussed in the theoretical framework. Appendix A provides no value of ALPHA for the first dataset, and it proposed (without any justification) a value of 1 for the other two domains. Appendix B is extremely brief and not very helpful.  The authors make no recommendation on how to tune alpha , and the argument that even the worst alpha (in the 0.1 - 0.9 range) is better than the original model is fairly weak, given the wide variations of the accuracy due to changes the value of alpha."**
>
> **RESPONSE:** We have revised the Appendix of the paper to provide detailed parameter $\alpha$ values used in each experiment (see Appendix A). In Appendix B, we also presented an ablation study with different $\alpha$ for the image classification task in Figure 6. A larger $\alpha$ will make the main task learning favors more on minimizing the reweighting loss rather than simply minimizing the difference between model predicted labels and the given labels $\tilde{Y}$. In our exeriment results in Figure 6, it actually suggests that with the $\alpha$ taking value from the recommended range $[0.1,0.9]$, SCL is not very sensitive to $\alpha$. The maximum difference of accuracy between the worst and best choice of $\alpha$ is only about $2\%$ under different missing rates. We believe that the insensitivity to $\alpha$ is actually a strength of SCL, since if an algorithm's performance is highly sensitive to model parameters or dependent on parameter tuning, it is a not very robust algorithm.

---

### Official Review · Reviewer_3FG6 · 2021-11-02

**Correctness:** 3
**Technical Novelty And Significance:** 3
**Empirical Novelty And Significance:** 3
**Recommendation:** 5
**Confidence:** 3

**Main Review:**

Strengths
- The proposed framework seems to be novel.
- It works well in experiments, on a wide range of tasks (classification, label propagation, and data imputation).
- It seems to be potentially beneficial for many domains, since it does not have domain-restrictions, while many previous SSL methods rely on certain image domain techniques such as consistency regularization (and data augmentation).

Weaknesses
- Since the proposed method is only compared with the original pseudo-label method, comparing with other extensions of pseudo-labelling methods that are mentioned in Section 5 will make the contributions more clear.
- In addition to the papers mentioned in Section 5, there are a few papers that try to address the error accumulation in semi-supervised learning methods that is observed in pseudo-labelling. For example: "In Defense of Pseudo-Labeling: An Uncertainty-Aware Pseudo-label Selection Framework for Semi-Supervised Learning" from ICLR 2021 and "Repetitive Reprediction Deep Decipher for Semi-Supervised Learning" from AAAI2020.

Questions
- I am not sure if I understood the experiments correctly. As the missing rate goes higher, do we have more unlabeled samples (as explained in the last paragraph of page 6), or do we have more noisy-labelled samples (as explained in 1st paragraph of Section 4.1)?
- Can we show the 3rd task (data imputation) in Figures 2 to 4?
- One of the benefits of the method seems to be that it can be incorporated into a wide range of SSL algorithms. I think the paper demonstrated that it can be used to enhance pseudo-labelling method, but what kind of other SSL algorithms can SCL incorporate?

Minor questions and comments
- SSL is a very hot topic and there has recently been many advances. Since the experiments do not compare with many of the recent works, it would be better to emphasize why they were not compared. (For example, Section 1 has a discussion on how recent SSL methods utilize consistency regularization, which relies on heavy data augmentation techniques that is only available in certain domains.)
- What kind of value for parameter alpha is used in the image classification? (For the other two tasks, I think the appendix explains that alpha is 1).
- If we are given a labeled dataset L and unlabeled dataset U, it seems we can automatically construct vector M (which is explained in end of page 2). If this is correct, then why do we need M as an input in Algorithm 1 in page 6?
- What is P introduced in the beginning of Section 2.2? It seems like it is a set from the $p \in P$ notation but since it compares with M in the loss function, it also looks like a vector.
- typo "perforamnce" in page 6
- Should $m_i, m_j$ in the beginning of page 3 be $M_i, M_j$?
- Is $Y$ a label space ($y \in Y$), or is it the full set of labels in the training dataset ($Y = Y_L \cup Y_U$)?
- Ideally it would be better to perform several trials and report mean/standard error in Table 1.

=========== after rebuttal

Thank you for answering my questions. The additional experiments are helpful to have a better understanding about the proposed method. It looks like the advatangeous points of the proposed method is now about the low computational costs, according to the new experiments including UPS, rather than better performance. Although this still may be beneficial for the research community, it seems to be slightly less significant and also may affect the storyline. I would like to also recommend to put the new experiments with UPS in the main paper instead of the appendix.

**Summary Of The Paper:**

The paper proposes a novel framework for semi-supervised learning, that solves two issues of previous methods: 1) over-reliance on labeled data and 2) error accumulation. It shows that jointly solving the main task together with another task (that discriminates whether the data label is real or not) leads to better performance.

**Summary Of The Review:**

The proposed method seems to have some nice benefits, but I feel there are a few weaknesses that should be addressed. I also have a few questions and it would be helpful if the authors can take a look at the previous section (main review).

---

> ### Author Response · Authors · 2021-11-21
> **Response to Reviewer 3FG6**
>
> **"SSL is a very hot topic and there has recently been many advances. Since the experiments do not compare with many of the recent works , it would be better to emphasize why they were not compared. (For example, Section 1 has a discussion on how recent SSL methods utilize consistency regularization, which relies on heavy data augmentation techniques that is only available in certain domains .)"**
>
> **RESPONSE:** We thank the suggestions provided by the reviewer. As we are proposing a general framework for SSL problems, hence we did not compare with those SSL algorithms that have domain restrictions, e.g. consistency regularization (relies on certain image domain techniques) and some pseudo-label based methods (which cannot be used for SSL problems on regression setting) in the previous version of the paper.
> However, in our revised version of the paper, we have added comparative experiment of SCL against some recent SOTA SSL algorithms. Please refer to our revised Experiments section and Appendix for detailed comparative results.
>
> **"What kind of value for parameter alpha is used in the image classification? (For the other two tasks, I think the appendix explains that alpha is 1 )."**
>
> **RESPONSE:** The alpha hyperparameter we used in the image classification task is 0.6. We also present the relationship of model accuracy and different $\alpha$ values in the ablation study in Figure 6 of the Appendix.
>
> **"If we are given a labeled dataset L and unlabeled dataset U, it seems we can automatically construct vector M (which is explained in end of page 2). If this is correct, then why do we need M as an input in Algorithm 1 in page 6?"**
>
> **RESPONSE:** We do not explicitly compute M in our algorithm, we can simply use the labeled/unlabeled information in SCL for computation. The use of $M$ in our formulation is mainly for notational convenience, as it is needed to formulate and compute the discriminator loss $\mathcal{L}_B$.
>
> **"What is P introduced in the beginning of Section 2.2? It seems like it is a set from the p∈P notation but since it compares with M in the loss function, it also looks like a vector."**
>
> **RESPONSE:** $P$ is the output vector of the discriminator, which has the same shape as M and represents the confidence measure of each sample, i.e. output value close to 1 if it is identified as labeled data, 0 otherwise.
>
> **"typo "perforamnce" in page 6 "**
>
> **RESPONSE:** Thanks to the reviewers for pointing out this typo, we have corrected it in the revised paper.
>
> **"Should mi,mj in the beginning of page 3 be Mi,Mj ?"**
>
> **RESPONSE:** We have revised $m_i, m_j$ to $M_i, M_j$ in our revised paper as recommended by the reviewer.
>
> **"Is Y a label space ($y\in Y$), or is it the full set of labels in the training dataset (Y=Y_L∪Y_U)? "**
>
> **RESPONSE:** $Y=Y_L∪Y_U$ is the full set of labels which consist of label for observed $Y_L$ (true labels) and unobserved $Y_U$ (random or pseudo-labeled labels) samples.
>
> **"Ideally it would be better to perform several trials and report mean/standard error in Table 1."**
>
> **RESPONSE:** We thank the reviewer for the comment. We present the average results over 10 random seeds for the experiments. We have added the mean/std of our experiment results in Table 1. Please refer to our revised paper for more details.

---

> ### Author Response · Authors · 2021-11-21
> **Response to Reviewer 3FG6**
>
> We thank the reviewer for the positive feedback. The reviewer is correct that our framework does not have domain-restrictions and can be potentially applied to a wide range of SSL problems. SCL only requires the main task model to have an explicitly defined loss function $loss_A(\cdot)$. We answer the remaining of reviewer's questions as follows:
>
> **"Since the proposed method is only compared with the original pseudo-label method, comparing with other extensions of pseudo-labelling methods  that are mentioned in Section 5 will make the contributions more clear."**
>
> **RESPONSE:** We have added additional supplementary experiments on CIFAR-10 dataset and compared SCL with several advanced pseudo-labeling methods, including the state-of-the-art method: uncertainty-aware pseudo-label selection (UPS) (Rizve et al. ICLR 2021). We have revised the experiment section as well as the Appendix to report these results. SCL offers very competitive performance while has much higher computational efficiency compared with the these SSL methods.
>
> **"In addition to the papers mentioned in Section 5, there are a few papers that try to address the error accumulation in semi-supervised learning methods that is observed in pseudo-labelling. For example: "In Defense of Pseudo-Labeling: An Uncertainty-Aware Pseudo-label Selection Framework for Semi-Supervised Learning " from ICLR 2021 and "Repetitive Reprediction Deep Decipher for Semi-Supervised Learning" from AAAI2020."**
>
> **RESPONSE:** We thank the reviewer's reminder. As discussed in our previous response, we have included the discussion of these papers in the related work section in the revised paper, and also added comparative experiments between our method and the method mentioned by the reviewer (e.g. UPS).
> These new results further demonstrate that although SCL addresses the error cumulation in a different way compared with the existing pseudo-labeling method, it is effective and achieves good performance.
>
> **"I am not sure if I understood the experiments correctly. As the missing rate goes higher, do we have more unlabeled samples (as explained in the last paragraph of page 6), or do we have more noisy-labelled samples (as explained in 1st paragraph of Section 4.1)?"**
>
> **RESPONSE:** We apologize for the vague description in the paper. In our experiments, higher missing rates correspond to more unlabeled samples. The reason that we fill the unlabeled samples with random labels as mentioned in Section 4.1 is for training initialization. SCL uses a different pseudo-labeling strategy as compared to most existing pseudo-labeling methods. SCL pseudo-labels all unlabeled samples and imposes soft-labeling weights based on discriminator outputs; while most pseudo-labeling methods only select and introduce a few reliable pseudo-labeled samples into the training set. Hence at the initial step of SCL, as the main task model is not yet trained, the first round of pseudo-labeling will simply fill all the unlabeled samples with random labels. Specifically, the unlabeled samples for classification tasks will be filled with a random data label; and for regression tasks, we fill them with the mean value of labeled data adding random noises.
> We have revised our paper to add more clarity.
>
> **"Can we show the 3rd task (data imputation) in Figures 2 to 4?"**
>
> **RESPONSE:** We thank the reviewer for this suggestion. We have added the learning curves and performance results for the data imputation task in Figure 2 and 3.
> As most pseudo-labeling methods are designed for classification tasks rather than regression tasks, the concept of pseudo-label and error accumulation are not well defined under regression setting. Hence, we focus on providing the results on classification tasks in Figure 4.
>
> **"One of the benefits of the method seems to be that it can be incorporated into a wide range of SSL algorithms. I think the paper demonstrated that it can be used to enhance pseudo-labelling method, but what kind of other SSL algorithms  can SCL incorporate?"**
>
> **RESPONSE:** As stated in our previous discussion, the only requirement of applying SCL on SSL algorithms is to have an explicitly defined loss function $loss_A(\cdot)$. This is generally not a issue for most SSL tasks. For some very special SSL problems where the model is learned with an implicit loss function or cannot explicitly define a loss function, the SCL framework may not be directly applicable.

---

### Author Response · Authors · 2021-11-21
**General Response**

**Regarding the comment of "lack of experiment against other advanced pseudo-labeling methods"**:

**RESPONSE:** We have added additional supplementary experiments on CIFAR-10 dataset and compared SCL with several advanced pseudo-labeling methods, including the state-of-the-art method: uncertainty-aware pseudo-label selection (UPS) (Rizve et al. ICLR 2021). We have revised the experiment section as well as the Appendix to report these results. SCL offers very competitive performance while has much higher computational efficiency compared with the these SSL methods.

**Regarding the comment of "comparing SCL with fully supervised baseline model"**:

**RESPONSE:** We have added experiments under 0% missing rates for the image classification task, and these results are added in Table 1. Supprisingly, we observe that in the case of 0% missing rate (fully supervised), SCL can achieve even higher accuracy compared with the fully supervised base model (accuracy: $86.2\pm 0.13\%$ (SCL) vs $83.85\pm 0.13\%$ (base model)). Under this case, both the main task model and the discriminator are learned with full data. The discriminator are trained with only label 1 (all labels are observed). The explaination for this might due to following reasons:
* The involvement of extra discriminator in a way improves the capacity of main task model $f$. It provides additional information to soft-weighting the losses for data samples, which can potentially beneficial for the main task model learning.
* The self-interested behavior of the main task $f$ on discriminator $d$ introduces some sense of adversarial learning in this process. The discriminator is actually learning implicitly by minimizing the maxima of functional $J(f)$ rather than simply minimize $\mathcal{L}_B$ that learn to ouput $p=1$. This forces the discriminator to focus more on mining the hidden pattern in the input data $X$, which can provide important supplementary information and meaningful reweighting loss $loss_W$ for the learning of main task $f$.

Moreover, SCL achieves similar level of accuracy as the fully supervised base model even with $60\%$ missing rate. These results suggests that SCL can also be used in supervised learning problems to boost model performance.

---

### Author Response · Authors · 2021-11-21
**Revision summary**

We thank the reviewer for all the constructive comments. In this revision, we have revised and added more results to address the reviewers' concerns. Including:
1. Additional supplementary experiments on CIFAR-10 dataset and compared SCL with several advanced pseudo-labeling methods, including the state-of-the-art method: uncertainty-aware pseudo-label selection (UPS) (Rizve et al. ICLR 2021). Detailed results are included in Appendix B.2.
2. Additional experiments under 0% (fully supervised), 90% missing rates for the image classification task. Detailed results are included in Tabel 1.
3. Revising the text of the Experiments section and the Appendix to provide more details about our model and experiment settings.
4. Adding the mean/standard error in Table 1 and Table 2.
5. Adding the convergence curves of data imputation task in Figure 2.
6. Adding the performance under different missing rate of data imputation task in Figure 3.
7. Adding horizontal lines with the accuracy of the base model in Figure 6.
8. Revising all the typos as pointed out by the reviewers.
9. Revising vague description of the notation in the paper to add more clarity.
10. Adding additional references to the related work section:
    - A principled end-to-end framework with a repetitive reprediction training strategy (R2-D2). "Repetitive Reprediction Deep Decipher for Semi-Supervised Learning" AAAI2020 https://arxiv.org/pdf/1908.04345.pdf
    - An uncertainty-aware pseudo-label selection (UPS) framework"In Defense of Pseudo-Labeling: An Uncertainty-Aware Pseudo-label Selection Framework for Semi-Supervised Learning" ICLR 2021 https://openreview.net/pdf?id=-ODN6SbiUU

---

### Decision · Program_Chairs · 2022-01-20

**Decision:**

Reject

**Comment:**

This paper proposes a new method for the important problem of semi-supervised learning. This method relies on an auxiliary task, label observability prediction, to weight the examples according to the confidence in their pseudo-labels, so as to avoid the propagation of errors encountered in self-training. Limited experiments show that the proposed method can compete with other methods in terms of performance or training time. On the positive side, all evaluators agree on the potential value of the proposed approach, which is generic in nature. On the negative side, the experimental evaluation, although strengthened during the discussion, is not yet strong enough to have really convinced of the real merits of the method. In particular, comparisons with the state of the art still need to be improved. In addition, the paper would benefit from some rewriting, in particular of the mathematics (e.g. the d notation for task B should be avoided as suggested by one reviewer, there is a misplaced partial derivative in equation 6). The authors could also simplify their derivation by using the envelope theorem. I therefore recommend rejection, with an encouragement to strengthen the experimental part, and to improve the derivation of the proposed method.